# Metacognitive biases in anxiety-depression and compulsivity extend across perception and memory

**Tricia X. F. Seow**[1,2]*, **Stephen. M. Fleming**[1,2,3], **Tobias U. Hauser**[1,2,4,5]

1 Max Planck UCL Centre for Computational Psychiatry and Ageing Research, Department of Imaging Neuroscience, University College London, London, United Kingdom, 2 Functional Imaging Laboratory, Department of Imaging Neuroscience, University College London, London, United Kingdom, 3 Department of Experimental Psychology, University College London, London, United Kingdom, 4 Department of Psychiatry and Psychotherapy, Faculty of Medicine, University of Tübingen, Tübingen, Germany, 5 German Center for Mental Health (DZPG), partner site Tübingen, Tübingen, Germany

* t.seow@ucl.ac.uk

## Abstract

Metacognitive biases are characteristic of common mental health disorders like depression and obsessive-compulsive disorder (OCD). However, recent transdiagnostic approaches consistently contradict traditional clinical studies, with overconfidence in perception among highly compulsive individuals versus underconfident memory in OCD patients. To reconcile these differences, we investigated whether these metacognitive divergences may arise due to cognitive domain-specific effects, comorbid overshadowing effects, and/or different manifestations at disparate levels of a local to global metacognitive hierarchy. Using a transdiagnostic individual differences approach with a general population sample (N = 327), we quantified metacognitive patterns across memory and perception. Across cognitive domains, we found that underconfidence was linked to anxiety-depression and overconfidence was linked to compulsivity. While both anxiety-depression and compulsivity were predominantly explained by global low self-esteem, other associations varied across a confidence hierarchy, with compulsivity exhibiting more specific alterations at more local metacognitive levels. Our results support a domain-general alteration of metacognition across mental health dimensions, with differential contributions from distinct levels of a metacognitive hierarchy.

## Introduction

Altered insight is a hallmark of many psychiatric disorders [1]. In experimental studies, quantifying metacognition provides one route towards studying the processes contributing to distorted insight [2], and consequently has been increasingly examined in clinical populations. Metacognition is regularly studied by asking people to evaluate their confidence in their performance in a particular domain, with "metacognitive bias" referring to cases in which these self-evaluations diverge from reality [3]. Empirically, this entails examining if there are biases in the summary statistics (e.g., mean) of trial-by-trial confidence ratings within a cognitive task where performance is equated across participants through a staircase procedure [3].

**Data availability statement:** Stimuli for the memory task is publicly accessible (https://osf.io/mey48/). The data and analysis code to reproduce the main analyses and figures are available at https://osf.io/jm2at/.

**Funding:** TXFS is a Sir Henry Wellcome Postdoctoral Fellow (224051/Z/21/Z) based at the Max Planck UCL Centre for Computational Psychiatry and Ageing Research. SMF is a CIFAR Fellow in the Brain, Mind & Consciousness Program, and is funded by UK Research and Innovation (UKRI) under the UK government's Horizon Europe funding guarantee (selected as ERC Consolidator, grant number 101043666). TUH is supported by a Sir Henry Dale Fellowship (211155/Z/18/Z; 211155/Z/18/B; 224051/Z/21) from Wellcome/ Royal Society, a grant from the Jacobs Foundation (2017-1261-04), the Medical Research Foundation, a 2018 NARSAD Young Investigator grant (27023) from the Brain & Behavior Research Foundation, and a Philip Leverhulme Prize from the Leverhulme Trust (PLP-2021-040). This research was funded in whole, or in part, by the Wellcome Trust (211155/Z/18/Z). The Max Planck UCL Centre for Computational Psychiatry and Ageing Research is a joint initiative supported by UCL and the Max Planck Society. The Wellcome Centre for Human Neuroimaging is supported by core funding from the Wellcome Trust (203147/Z/16/Z). For the purpose of Open Access, the author has applied a CC BY public copyright licence to any Author Accepted Manuscript version arising from this submission. Funders did not play any role in the study design, data collection and analysis, decision to publish, or preparation of the manuscript.

**Competing interests:** TUH consults for limbic ltd and holds shares in the company, which is entirely unrelated to the current project. The other authors declare no conflicts of interest.

Several patient studies have found metacognitive biases across multiple psychiatric disorders [4]. For instance, patients with depression [5–7] and obsessive-compulsive disorder (OCD) [8] consistently express lowered confidence in their decisions/actions. The latter disorder is even cast as a 'disorder of doubt' [9,10] and lowered confidence was found across various memory paradigms [11–16] over and above impaired memory performance [8]. However, recent studies have observed more differentiated effects when adopting a transdiagnostic approach with general population samples in metacognitive research using perceptual or reinforcement learning decision tasks [17]. These studies, including our own, utilised an established set of latent factors of psychopathology encompassing 'anxiety-depression', 'compulsivity and intrusive thoughts' (hereafter: 'compulsivity'), and 'social withdrawal' dimensions, derived using factor analysis [18]. Notably, clinical samples also produce these three transdiagnostic dimensions [19]. In particular, the 'compulsivity' construct not only includes compulsive behaviours characteristic of OCD, but also encompasses other compulsive-type symptoms such as addiction and eating disorders [18]. These features have been shown to have greater biological validity in relation to known OCD cognitive deficits than diagnosis itself [20]. In probing the factors' associations with metacognition, these studies report that individuals with high 'anxiety-depression' scores express metacognitive underconfidence, whilst individuals with high 'compulsivity' scores express overconfidence [21–26]. Taken together, these findings suggest a striking and hitherto unexplained dissociation in which a diagnosis of OCD is linked to underconfidence using memory paradigms, whereas transdiagnostic compulsivity is linked to overconfidence in other cognitive tasks beyond memory.

The reasons for these opposing findings are unclear, with a range of explanations on offer. First, metacognition may only partially generalise across cognitive domains [27–32] meaning that poor metacognition in memory tasks does not necessarily predict poor metacognition in perceptual decision tasks (termed domain-generality). If they manifest differently, it might explain the findings of underconfidence in one domain, and overconfidence in another. While there is support for metacognitive domain-generality in perception and semantic knowledge manifesting in mental health [21], the metacognitive domain-generality of perception and episodic/working memory in compulsivity is untested and key to the current question. Second, different findings may arise due a mixture of psychopathological traits across the different samples [33]. Due to high co-morbidity across disorders (e.g., depression in OCD patients), the effect of one symptom dimension could overshadow or mimic the effect of another, i.e., OCD patients may exhibit lowered confidence due to their co-morbid depression, rather than their primary OCD symptoms [25,34]. Third, we and others have recently shown that the expression of confidence may reflect an internal hierarchy, ranging from local confidence in single decisions to higher-level global assessments such as self-esteem [22,35,36]. Metacognitive biases across different psychopathologies could manifest at different levels of a confidence hierarchy, leading to apparent dissociations between different studies [17].

Here, we set out to disentangle the conflicting findings relating to metacognitive biases found in compulsivity and OCD. We focused on using a transdiagnostic individual differences approach with a general population (but not explicit patient) sample, and tested metacognition across the cognitive domains of perception and memory, and across a hierarchy of metacognitive measures. We quantified six measures spanning local to global metacognition—from local (trial-by-trial) task confidence ratings, to global measures of pre- and post-task evaluations of performance, to even more global estimates of cognitive domain ability, to the highest-level global metacognitive measures of self-esteem and self-efficacy. We then investigated the involvement of these different hierarchical levels of metacognition by probing their contributions to dimensions of anxiety-depression and compulsivity. Our results (N = 327) show a domain-general effect of low confidence linked to anxiety-depression and

high confidence linked to compulsivity, suggesting that the previously observed differences of memory underconfidence in OCD versus perceptual overconfidence in compulsivity are unlikely to arise due to domain-specific metacognitive differences, and are more likely due to overshadowing effects. Moreover, we show that mental health dimensions are linked to metacognitive impairments at distinct levels of a metacognitive hierarchy—with predominant contributions of low global self-esteem to both anxiety-depression and compulsivity, but with additional bi-valenced multi-hierarchical contributions of confidence to compulsivity, including low global post-task confidence and self-esteem together with high self-efficacy and local task confidence.

## Materials and methods

### Ethics statement

We obtained and performed in accordance to the ethical approval for the study procedures by UCL's Research Ethics Committee (15301/001).

### Participants

N = 400 participants were recruited online via the worker platform Prolific (https://www.prolific.co/) from 7 June 2023 to 19 July 2023. All participants were 18–55 years old, currently residing in the United Kingdom, fluent in English, have normal or corrected-to-normal vision, and have at least 90% approval rate on Prolific. They could only access the experimental task if they provided informed consent online by checking all the tick boxes of the consent form and clicking a 'complete' button. For reimbursement, they received £8.26/hour plus a bonus of up to £4 based on their performance.

### Power analysis

A power analysis was conducted using the effect size from a previous study examining associations between perceptual local confidence alterations and transdiagnostic dimensions of anxiety-depression, compulsivity and intrusive thought ('compulsivity') and social withdrawal [26]. The prior study consisting of 210 task trials reported a positive association of anxiety-depression, and negative association of compulsivity, with mean confidence level in a regression model (Cohen $f^2 = 0.107$). Given that the current tasks consisted of 150 trials each, we re-estimated the prior model for 150 trials (Cohen $f^2 = 0.0948$). We then used this effect size to calculate sample size, which suggested that N = 327 participants were required to achieve 90% power at 0.001 significance. We estimated N = 400 as the target number of participants to reach the necessary sample size after accounting for expected data exclusions. In a post-hoc analysis, the final sample size at N = 327 was also sufficiently powered (80% power, 0.05 significance) for the current regression models to detect the expected effect sizes (above Cohen $f^2 = 0.0557$).

### Exclusion criteria

To ensure data quality, multiple attention and comprehension checks were integrated into the study. For the experimental tasks, participants were required to get 100% correct on a short comprehension quiz on the task instructions before they were allowed to start either of the main tasks. Should they fail, they were directed back to the instructions before another quiz attempt. If they failed on their 3rd attempt, they were brought back to the very beginning to complete the practice trials again before another quiz attempt. For the questionnaires, the battery set included two attention check question items requiring participants to select a specific answer.

Participants were excluded from the analysis if they:

- Had incomplete datasets, arising from mishaps in remote data collection (e.g., portions not saved) (N = 18)

- Had > 50 attempts to identify all the stimuli in the stimuli recognition training portion for the metamemory task (See Metamemory Task) (N = 0)

- Had > 5 attempts of the comprehension quiz (either task) (N = 0)

- Failed at least one attention check question in the questionnaire battery (N = 10)

- Had r > 0.5 correlation of initial default confidence rating with final confidence rating across trials in tasks (N = 1 (perception))

- Had task accuracy that diverged from expected of staircase ( < 60% or > 85%) (N = 6 (memory), N = 1 (perception))

- Had same trial-by-trial confidence rating > 80% of the trials (N = 3 (memory), N = 3 (perception))

- Chose the left/right option > 80% the time (N = 3 (perception))

- Recorded stimulus presentation time deviating > 50 ms from expected (N = 33 (perception))

- Reported a task stimulus presentation error (N = 1 (memory))

- Mean confidence reaction time > 20 s (N = 1 (memory))

In total, N = 73 (18.25%) were excluded, leaving N = 327 participants (136 female (41.59%), 1 other gender (1.22%)) for analysis.

We additionally excluded trials with implausibly slow reaction times of > 10 s and/or ± 3 standard deviations from the per-subject mean for each participant. 1.59% (metamemory task) and 1.82% (metaperception task) of all trials were excluded.

## Procedure

We employed a randomised, cross-over within-subject study design (Fig 1A). We first assessed cognitive self-ability metacognition before each participant completed two experimental task sets assessing memory or perception (Fig 1B and 1C). Each of these sets included a tutorial with practice trials, pre-task global metacognitive ratings, the main experimental task with trial-by-trial confidence ratings, followed by post-task global metacognitive ratings. After the task sets, we asked participants about their insight in how their confidence evaluations were related between the experimental tasks. Finally, self-report questionnaires measuring individual differences in demographics, psychopathology, self-beliefs and intellectual abilities (IQ) were administered.

In total, six metacognitive measures across the metacognitive hierarchy were assessed (Fig 1D). These included local task confidence ratings (Local task metacognition), global pre- and post- task evaluations (Global task metacognition), even more global cognitive ability self-performance estimates (Self-ability metacognition/estimates), and finally the most global measures of self-esteem and self-efficacy (Questionnaires). Refer to each section for details of their quantification.

The entire study was programmed in React v.18.2.0 (https://react.dev/) (JavaScript library), developed in an app bootstrapped by Create React App (https://github.com/facebook/create-react-app) and hosted on Scalingo (https://scalingo.com/).

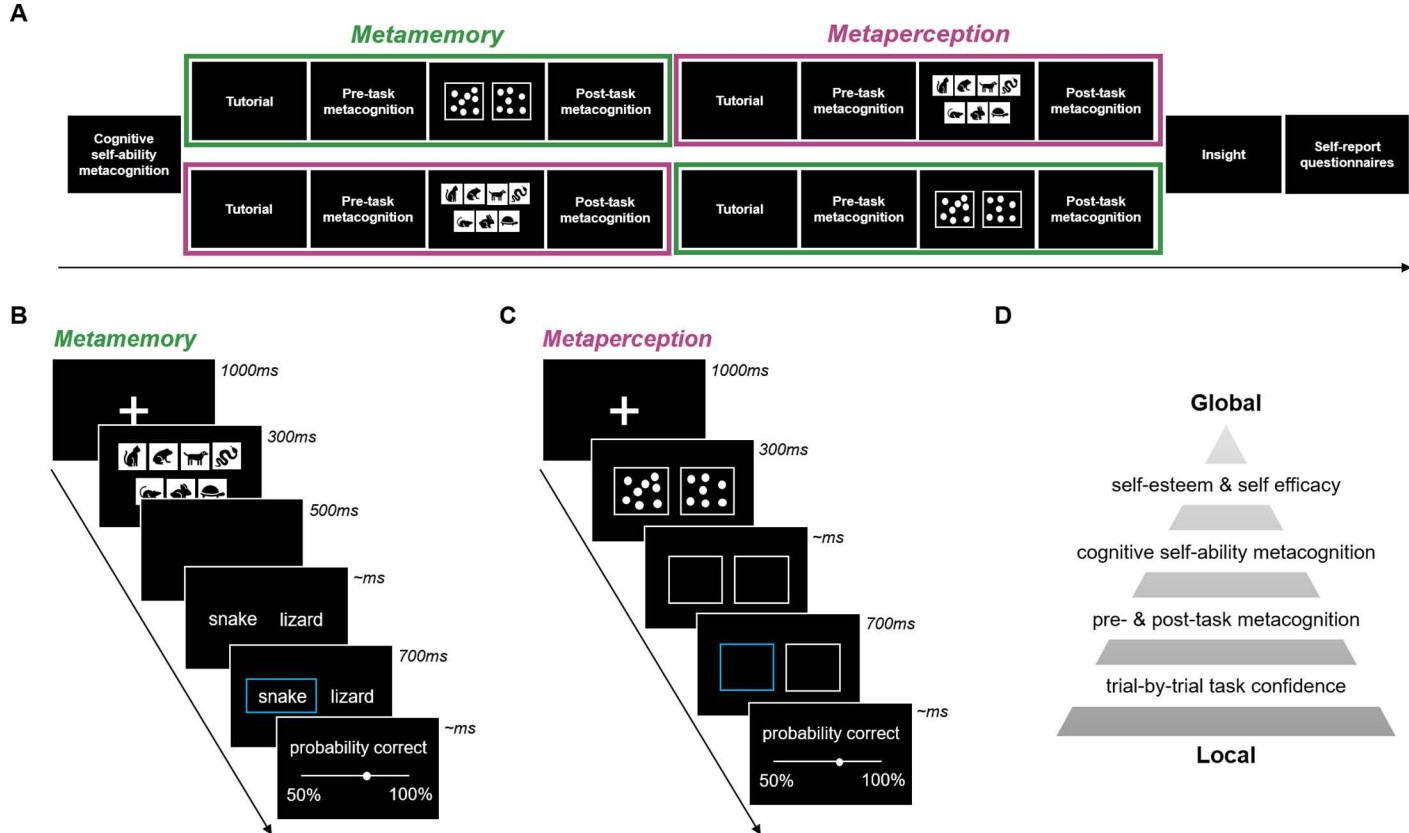

Fig 1. Experimental task schedule and paradigms. (A) Task schedule. Participants were randomly allocated to complete either the metamemory or metaperception task set first. (B) Metamemory task. Participants were to select the word of the stimuli that was previously displayed amongst an array. (C) Metaperception task. Participants were to select the box which displayed more dots. In both (B) and (C), confidence ratings of the decisions were made on a continuous scale (50%–100% probability correct) for every trial. (D) Metacognitive measures. We quantified six different measures spanning local to global estimates across the metacognitive hierarchy. (Mean of) Trial-by-trial confidence ratings were considered the most "local" as they were reflections of a single decisions, while pre- and post-task metacognition were more "global" entailing reflections about an entire task. Even more global cognitive self-ability metacognition captured the evaluation of an entire cognitive domain, whilst the most global levels of self-esteem or self-efficacy measures reflect a combination of cognitive self-evaluations with other trait influences.

## Self-ability metacognition/estimates

Directly after the consent pages and prior to the experimental task sets, we asked participants to rate their abilities of memory and perception on a 1–10 scale, from 'worse than everyone' to 'better than everyone':

- How you do generally rate your memory ability? For example, in remembering events that you experienced a long time ago.

- How you do generally rate your perception ability? For example, how good you are at spotting hidden things, like birds in a forest.

## Experimental tasks

Participants then began with one of the experimental task sets by random assignment in the order of which was performed first (N = 159 (48.62%) memory first, N = 168 (51.38%) perception first). We employed two-alternative forced choice memory and perceptual

decision-making tasks with trial-by-trial confidence ratings (Fig 1B and 1C). Both tasks were presented with a cover story. Participants were told that they boarded our spaceship to help out after an asteroid hit and damaged the spaceship. In the metamemory task, participants were to memorise sets of animal images and then were presented with two words (that each described an animal). They had to remember and select which animal was seen previously to help allocate them into their pods on the spaceship. In the metaperception task, participants were to distinguish high charged batteries (box with more dots) to help power up the spaceship. In both tasks, their confidence in their choice (higher charged battery/animal that was seen) was rated after every trial on a continuous 50–100% probability correct scale. Each task set consisted of a tutorial with 25 trials, plus the main task of 150 trials split over 3 blocks. Performance in both tasks were controlled by a two-down one-up staircase procedure (targeting ~71%) that was initiated during the 25 practice trials to minimize burn-in period (S1 File).

## Metamemory task

On each trial, participants were presented with a fixation cross (1000 ms), followed by a presentation of a set of animal stimuli (1000 ms) before the screen was cleared (500 ms) (Fig 1B). Participants were then shown two words, one which described one of the stimuli in the presented set and one that did not. They were to select the word representing the stimuli that was shown previously. The chosen word was then outlined in blue (700 ms) before they rated their confidence in the decision. The staircase step-size was ± 1 stimulus for the entire task.

**Stimuli recognition task.** As part of the tutorial, participants were shown each of the task stimuli one at a time, and asked to select the word that describes the stimulus versus another word that did not. The correct word for all stimuli had to be chosen to move on to the main task instructions and practice trials. This was to prevent misinterpretation of what the stimuli were depicting.

**Harm versus non-harm stimuli.** We selected an array of 15 animal pictures as the stimuli set. They were in greyscale and matched in luminesce. Items in 8 of these were perceived as highly harmful while the other 7 were rated to have low harm levels in an affective rating study [37]. We indexed these stimuli into two groups (Harm, Non-Harm). See S2 File for the stimuli and analyses regarding the Harm versus Non-Harm stimuli effect on memory accuracy and confidence.

## Metaperception task

On each trial, participants were presented with a fixation cross (1000 ms) followed by a quick presentation of two boxes filled with differing number of dots (300 ms) (Fig 1C). One box would always have half the number of dots that could fill the box (313 dots out of 625 positions), whilst the other would have more than half (an increment of 1–312 dots). Empty boxes were then left on the screen, where participants were to select which box had the higher number of dots. The selected box was then outlined in blue (700 ms) before confidence ratings were given. Staircase step-size was calculated in log-space as implemented in prior studies [26], with a starting point of 4.2 (+ 70 dots), changing by ± 0.4 for the first 5 trials, ± 0.2 for the next 5 trials and ± 0.1 for the rest of the task. Dots were drawn on a trial-by-trial basis with react-konva v18.2.3 (https://konvajs.org/docs/react/index.html).

## Local task metacognition

We estimated local confidence (metacognitive bias) measure by the mean of the trial-by-trial confidence ratings over each task. Split-half reliability was high (r = 0.99) for both tasks. We also ran both individual and hierarchical Bayesian meta-d' analyses (HMeta-d) [38,39] to

obtain individual estimates of d' (sensitivity), meta-d' (metacognitive sensitivity) and m-ratio (meta-d'/d'; metacognitive efficiency), as well as group level estimates of m-ratio. Results of meta-d' model parameter analyses are reported in S3 File.

## Global task metacognition

Pre- and post- task self-performance estimates were assessed before and after each of main tasks respectively, by asking how many trials participants think they would get/have gotten correct with a 0–150 correct trials scale.

 Metamemory:

- Pre-task: Before we begin, out of 150 sets of animals, how many times you do think you will be able to select the correct animal seen in the set?

- Post-task: After going through all the 150 sets of animals, how many times do you think you selected the animal seen in the sets correctly?

 Metaperception:

- Pre-task: Before we begin, out of 150 set pairs of battery cards, how many times do you think you will choose the higher charge battery card correctly?

- Post-task: After going through all the 150 set pairs of battery cards, how many times do you think you selected all the higher charge battery cards correctly?

## Insight ratings

After completion of both task sets, we asked three insight questions:

- Did you prefer to complete the first task over the second task? (1–5 scale; 1: definitely not, 3: no difference, 5: definitely yes)

- How much did you feel that your confidence changed from completing the first task to finishing the second task? (1–7 scale; 1: decreased a lot, 4: no change, 7: increased a lot)

- How much did you feel that your confidence in the first task influenced your confidence on the second task? (1–5 scale; 1: not at all, 3: somewhat, 5: a lot)

## Questionnaires

In the final section, participants completed a battery of 11 self-report questionnaires measuring psychopathology and self-beliefs, the order of which was fully randomised.

 We assessed 9 mental health questionnaires to subsequently define transdiagnostic dimensions [18], which included:

- Alcohol addiction, using the Alcohol Use Disorder Identification Test (AUDIT) [40]

- Apathy, using the Apathy Evaluation Scale (AES) [41]

- Depression, using the Self-Rating Depression Scale (SDS) [42]

- Eating disorders, using the Eating Attitudes Test (EAT-26) [43]

- Impulsivity, using the Barratt Impulsivity Scale (BIS-11) [44]

- Obsessive-compulsive disorder (OCD), using the Obsessive-Compulsive Inventory - Revised (OCI-R) [45]

- Schizotypy, using the Short Scales for Measuring Schizotypy (SSMS) [46]

- Social anxiety, using the Liebowitz Social Anxiety Scale (LSAS) [47]

- Trait Anxiety, using the trait portion of the State-Trait Anxiety Inventory (STAI) [48]

    We also collected 2 self-belief metrics as a form of global metacognition measures:

- Self-esteem, using the Rosenberg Self-esteem Scale (RSE) [49]

- Self-efficacy, using the General Self-efficacy Scale (GSE) [50]

    Finally, we administered a short IQ evaluation using the International Cognitive Ability Resource (I-CAR) sample test [51].

## Transdiagnostic dimensions

To quantify mental health dimensions in our sample, we applied a previously defined transdiagnostic definition based on a weighted combination of items drawn from the 9 mental health questionnaires to elucidate three transdiagnostic dimensions of 'anxiety-depression', 'compulsivity', and 'social withdrawal' [18]. The prior study reported that the highest average item loadings came from the trait anxiety, apathy and depression questionnaires for the anxiety-depression factor, from the eating disorders, addiction and OCD questionnaires for the compulsivity factor, and from the social anxiety questionnaire for the social withdrawal factor.

    We used weights derived from the previous study with N = 1,413 to transform our scores as our sample size had a lower a subject-to-variable ratio (N = 327) for de novo factor analysis. Consistent with prior literature, the resulting transformed dimension scores of anxiety-depression, compulsivity and social withdrawal were moderately intercorrelated (r = 0.30– 0.53). To note, in the prior study, item 13 on the SDS was mistakenly phrased "I am restless and can't sleep" rather than "I am restless and can't keep still", but subsequent work have demonstrated the stability of the factor structure in new data, with and without this error [25,26]. Likewise, applying de novo factor analysis to our current sample replicated the same factor structure. See S4 File for the distribution and correlations of questionnaire and dimensional scores, and the de novo analysis.

## Analyses

All analyses were conducted in R version 3.6.0 via RStudio version 1.2.1335 (http://cran. us.r-project.org) and MATLAB 2019b (https://www.mathworks.com/). In R, we utilised the t.test() function (stats package) for two-sided paired t-tests, lm() function (lme4 package) for general linear modelling, cor.test() function (stats package) for Pearson's and Spearman's correlation tests, and the step() function (stats package) for stepwise regressions. Repeated cross validation analyses on model regressions were performed in MATLAB using fitglm(), predict() and immse() functions. We outline further detail of the main analyses below:

## General linear modelling

We asked if mean local confidence (Confidence) was associated with psychiatric questionnaire (Questionnaire) or dimension (anxiety-depression (AD), compulsivity (CIT), social withdrawal (SW)) severity with a general linear model where the Task Order (memory:1, perception:-1), age, gender, IQ, and their interaction with Task Domain (memory:1, perception:-1) were taken as co-variates. Confidence, questionnaire/dimension scores, age and IQ were z-scored. The models were:

    Confidence ~ (Questionnaire + Task Order + Age + Gender + IQ)*Task Domain
    Confidence ~ (AD + CIT + SW + Task Order + Age + Gender + IQ)*Task Domain

The main effect of Questionnaire or AD/CIT/SW (Dimension) on Confidence was interpreted as a general confidence effect with psychopathology. The interaction effect of Dimension*Task Domain was interpreted as the change of dimension score on confidence effect compared between memory versus perception.

We also assessed if global levels of metacognition (Global Metacognition) were associated with dimension (Dimension) scores. These included pre-task metacognition, post-task metacognition, self-performance estimates, self-esteem and self-efficacy scores. We utilised the same linear model as above, replacing the dependent variable with each global metacognition construct of interest. For pre-task metacognition, post-task metacognition and self-performance estimates, we estimated a score each for memory and perception. Thus, the models were:

Global Metacognition ~ (AD + CIT + SW + Task Order + Age + Gender + IQ)*Task Domain

For self-esteem and self-efficacy, as there was only one score per participant (no domain dependency), the models were:

Global Metacognition ~ (AD + CIT + SW + Task Order + Age + Gender + IQ)

The main effect of AD/CIT/SW (Dimension) on Global Metacognition was interpreted as a general metacognitive effect of psychopathology.

### Stepwise regression

We explored the importance of each metacognitive metric in contributing to dimension severity. With a base model of age, IQ and gender, we set the full model for forward iteration as:

Dimension ~ (Local Confidence + Pre-task Metacognition + Post-task Metacognition + Self-ability Metacognition)*Task Domain + Self-Esteem + Self-Efficacy + Age + IQ + Gender

All metacognitive regressors, age and IQ were z-scored.

### Repeated cross validation

To validate the models revealed by stepwise regression, we conducted 5-fold cross validation on 16 variations of these regression models (S5 File). We used cvpartition() with K-fold = 5, running fitglm() on the training set and predicting dimension scores with predict() on the test set. immse() was used to obtain root mean squared error (RMSE) of the model fits.

## Results

To evaluate metacognitive biases across hierarchies and domains, we conducted a large-scale (N = 327) online study with a general population sample. Each participant completed, in a randomly assigned order, a memory and a perceptual task designed to measure performance and confidence in these cognitive domains (Fig 1). On each trial, participants performed a two-alternative forced-choice (stimulus identification in previously shown picture array (memory); numerosity discrimination (perception)) before rating their confidence in their decision on a continuous confidence scale. Performances for both tasks were equalised using a staircase procedure (S1 File). Thereafter, participants completed a battery of self-report questionnaires assessing a range of psychopathological subclinical traits, which were transformed into transdiagnostic dimension scores encompassing 'anxiety-depression', 'compulsive behaviour and intrusive thought' (compulsivity) and 'social withdrawal' previously defined in prior transdiagnostic work [18,26].

### Local task confidence is closely correlated across perception and memory

We first tested if there was a domain-general relationship between confidence across the two tasks. To ensure that confidence levels were not driven by differences in task performance, we

examined task accuracies to ensure that the staircase procedures were successful in equating performance between individuals and tasks. Reassuringly, accuracy was well-calibrated for both memory (mean (M) = 71.40% correct, standard deviation (SD) = 2.22%) and perception (M = 71.85% correct, SD = 2.36%), with no significant difference between tasks (t(325) = 0.57, p = 0.57, 95% Confidence Interval (CI) = [−0.08 0.14]) (Fig 2A). As accuracy was titrated, task difficulty (estimated as the titrated average dot numerosity difference (metaperception) or average stimuli number of picture array shown (metamemory)) reflected the participants' endogenous perceptual/working-memory ability. Higher difficulty was indexed by smaller (log) dots difference (M = 3.20, SD = 0.46) in the metaperception task and higher number of stimuli (M = 7.94, SD = 1.81) in the metamemory task (Fig 2B). We found that average task difficulty was correlated across tasks (r = −0.28, p < 0.001, 95% CI = [−0.38 −0.18]), indicating that participants who achieved a higher difficulty level at one task also did so in the other.

Given that task accuracies were similar between participants, we next examined confidence for the two tasks by estimating the mean trial-by-trial confidence level as a local measure of metacognitive bias. We found that participants were overall slightly more confident (t(325) = 2.79, p = 0.006, 95% CI = [0.29 1.66]) in the metaperception task (M = 76.70, SD = 8.19) as compared to the metamemory task (M = 75.73, SD = 6.51) (Fig 2C). Mean confidence ratings were highly positively correlated across tasks (r = 0.65, p < 0.001, 95% CI = [0.59 0.71]), and remained significantly correlated even after adjusting for task difficulty (r = 0.48, p < 0.001, 95% CI = [0.39 0.56]), supporting a domain-general pattern of confidence.

We noted that correct trials had a higher mean confidence rating than incorrect trials for both memory (M = 15.83, SD = 5.39) and perception (M = 5.69, SD = 4.08) (Fig 2D), indicating that participants effectively used their performance to inform their confidence ratings. We also saw a positive correlation between tasks of the difference between correct and incorrect trial confidence (r = 0.17, p = 0.002, 95% CI = [0.06 0.27]), suggesting a domain-general relationship between metacognitive sensitivity to performance as well. However, this was only

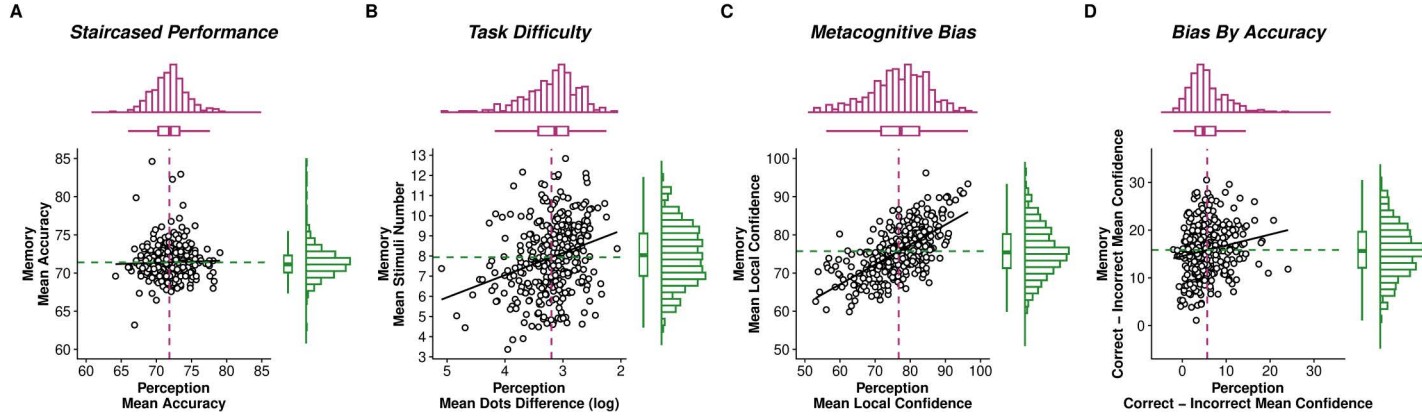

**Fig 2. Performance and confidence distributions from metaperceptual and metamemory tasks. (A) Task accuracy.** Both tasks reported similar levels of accuracy controlled by staircase procedures. **(B) Task difficulty.** As participants' accuracies were titrated, the difficulty at which the participants performed the task is another indicator of performance. Metaperception task difficulty level was indicated by the mean (log) dot difference between the stimuli (smaller difference = more difficult), while metamemory task difficulty level was indicated by the mean number of stimuli in the array during presentation time (more stimuli = more difficult), over the course of the experiment. **(C) Task confidence level.** Local metacognitive bias was estimated by the mean trial-by-trial confidence ratings of each task. Bias was positively correlated between tasks, suggesting a domain-general effect of confidence. **(D) Task confidence difference between correct and incorrect trials.** Correct trials garner higher confidence ratings than incorrect trials, indicating metacognitive awareness of performance. For hierarchical Bayesian modelling of metacognitive efficiency, see S3 File. For **(A)**, axes are in percentages, while for **(B)** and **(C)**, axes are in percentage of probability correct. Each circle represents an individual participant, dot lines represent means, diagonal lines indicate linear relationships. Histograms on the top and right insets of each figure represent the distribution of participants for each variable.

cautiously supported by a weakly positive cross-task correlation (mean group-level covariance ρ = 0.11, 95% highest density interval (HDI) = [−0.10 0.31]) in metacognitive efficiency (meta-d'/d'; the ability to distinguish between correct and incorrect judgments with confidence ratings) estimated within a hierarchical Bayesian model (HMeta-d [38]; S3 File).

## Anxiety-depression is associated with underconfidence and compulsivity is associated with overconfidence

We next asked whether previously observed relationships between confidence biases and psychopathology traits extended across domains. We first assessed the association between mean confidence level and each self-report questionnaire score we collected, adjusting for the influence of task domain, task completion order, age, IQ and gender. Echoing prior work in perception [26] and semantic knowledge [21] using the same set of questionnaires, we observed that several questionnaires, particularly subclinical traits that are central to disorders with low mood (e.g., apathy, depression, social anxiety, trait anxiety), were linked to lower confidence levels (β < −0.11, SE < 0.04, p < 0.005) across perception and memory (Fig 3A). We then refactored our questionnaire scores into previously defined mental health dimension scores (see Methods) [18], quantifying

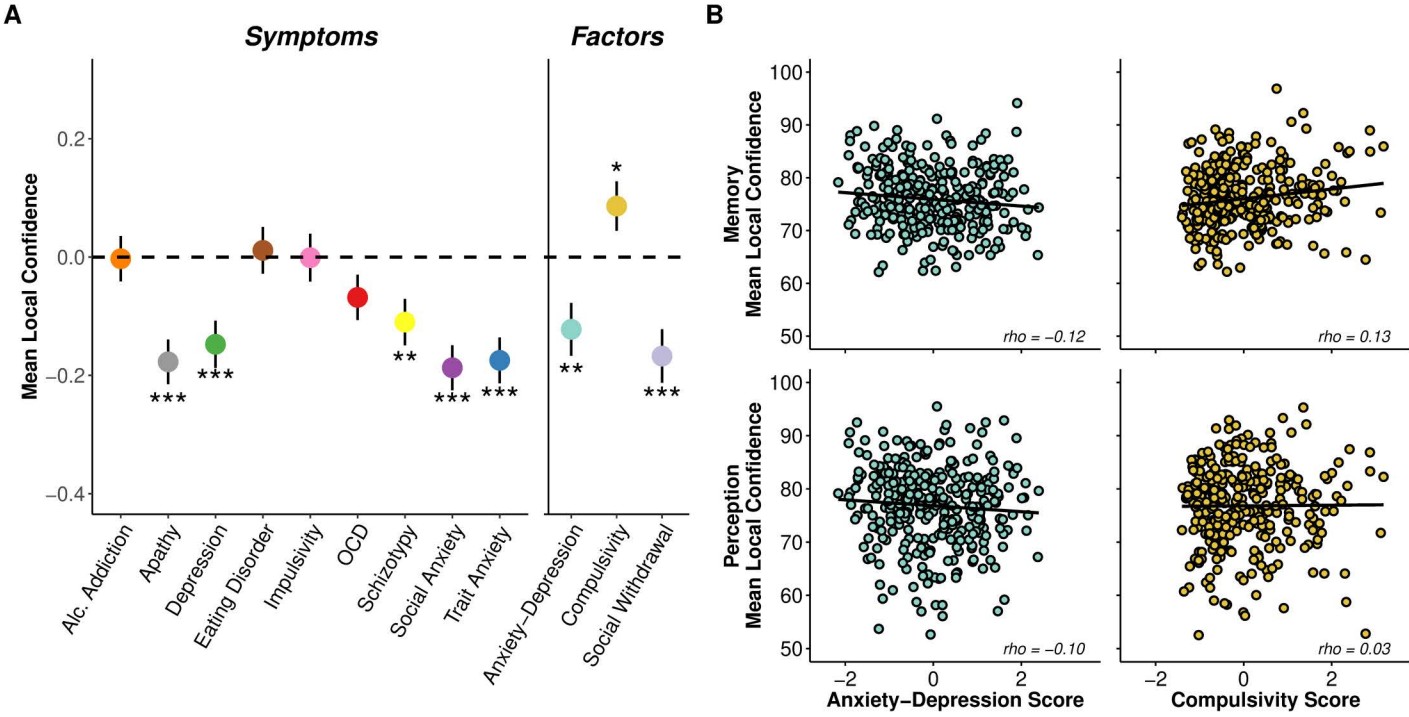

**Fig 3. Performance and confidence distributions from metaperceptual and metamemory task.** **(A)** Regression predicting mean local confidence from questionnaire and dimension scores. Higher apathy, depression, schizotypy, social anxiety and trait anxiety scores were significantly linked to lower confidence levels. When we refactored the questionnaires scores into the transdiagnostic dimension scores, lower confidence was linked to anxiety-depression and social withdrawal, while higher confidence was linked to compulsivity. Each questionnaire score was tested in a separate linear regression model, while dimension scores were included in the same model. The y-axis shows the z-scored change in mean confidence level as a function of 1 standard deviation increase of z-scored questionnaire/dimension scores. Error bars denote standard errors. *p < 0.05, **p < 0.01, ***p < 0.001. **(B)** Relationship between confidence and anxiety-depression or compulsivity in metaperception or metamemory. The relation of anxiety-depression to lower confidence was similar between memory and perception, while for compulsivity, its relation to higher confidence was stronger in memory than perception. For illustrative purposes, we obtained the confidence residuals for each participant after adjusting for anxiety-depression/compulsivity (anxiety-depression if compulsivity and confidence was to be examined, and vice versa), social withdrawal, task order, age, IQ and gender, separately for each task. We then correlated the confidence residuals and anxiety-depression or compulsivity scores with Spearman's correlation. Each circle indicates the confidence (measured on a 50–100% probability correct scale) residuals and dimension score for each participant.

anxiety-depression, compulsivity and social withdrawal levels for each participant. We observed specific bi-valenced relationships between confidence and the dimension scores —individuals high in anxiety-depression exhibited lower confidence ($\beta = -0.12$, SE = 0.04, p = 0.006) and individuals high in compulsivity exhibited higher confidence ($\beta = 0.09$, SE = 0.04, p = 0.04; Fig 3A).

To test whether these relationships between psychopathology and confidence differed between memory and perception, we investigated the interaction between task domain and mental health dimensions. We found no significant interaction between task domain and anxiety-depression ($\beta = -0.01$, SE = 0.04, p = 0.81), suggesting that the negative relationship between anxiety-depression and confidence is similar across perception and memory tasks (Fig 3B). However, there was a trending interaction between task domain and compulsivity ($\beta = 0.08$, SE = 0.04, p = 0.07), where individual differences in compulsivity showed a stronger positive association with higher confidence in memory than perception. The stronger association in memory than perception was also illustrated in task-specific regressions (S6 File).

Surprisingly, we also observed an association between social withdrawal and lower confidence ($\beta = -0.17$, SE = 0.05, p < 0.001) (Fig 3A). No prior study has reported such an association. There was no significant interaction effect between task domain and social withdrawal ($\beta = 0.03$, SE = 0.04, p = 0.56), meaning that lowered confidence was present across the tasks. Finally, to ensure that these effects were not driven by task performance, we established that task accuracy (ps > 0.44) and difficulty (ps > 0.16) did not show significant relationships with symptom dimensions. Overall, our findings thus suggest that relationships between mental health dimensions and local task confidence are relatively similar across task domains (albeit a stronger association with memory than perception), supporting a domain-general relationship between metacognition and psychopathology.

## Global metacognitive contributions to anxiety-depression and compulsivity

Beyond trial-by-trial local confidence ratings, we and others have conceptualised metacognition operating across different hierarchical levels [17]. To assess links between these levels and psychopathology, we also obtained more global metrics of task-specific metacognition in the form of pre-task and post-task self-performance estimates (see Methods). We also asked participants to introspect on their global perceptual and memory abilities prior to completing the tasks ("self-ability estimates"; e.g., "how good do you think your memory/perceptual ability is?"), and complete self-report questionnaires to assess more general self-beliefs including self-esteem and domain-general self-efficacy [49,50]. We then tested if these global metacognitive estimates differed as a function of dimension scores, adjusting for task domain and completion order (if relevant), age, IQ and gender.

We found no significant relationship between any of the dimension scores and pre- ($\beta$s > $-0.07$, SEs < 0.06, ps > 0.09) or post-task ($\beta$s > $-0.15$, SEs < 0.05, ps > 0.18) global metacognition (Fig 4A). However, we did find that individuals high in anxiety-depression reported lower self-ability estimates ($\beta = -0.21$, SE = 0.05, p < 0.001), in the absence of relationships with compulsivity ($\beta = 0.07$, SE = 0.05, p = 0.11) or social withdrawal ($\beta = -0.05$, SE = 0.05, p = 0.25). There was no interaction effect between dimension scores and cognitive domain on self-ability estimates (ps > 0.78), suggesting that anxiety-depression was equally linked to lower self-evaluation of memory and perception abilities. For self-esteem, both anxiety-depression ($\beta = -0.73$, SE = 0.03, p < 0.001) and social withdrawal ($\beta = -0.21$, SE = 0.03, p < 0.001) were linked to lower self-esteem, in the absence of relationships with compulsivity ($\beta = -0.03$, SE = 0.03, p = 0.27). For self-efficacy, anxiety-depression ($\beta = -0.53$, SE = 0.05, p < 0.001) and social withdrawal ($\beta = -0.25$, SE = 0.05, p < 0.001) were similarly linked to lower self-efficacy scores while compulsivity was linked to higher scores ($\beta = 0.12$, SE = 0.05, p = 0.009).

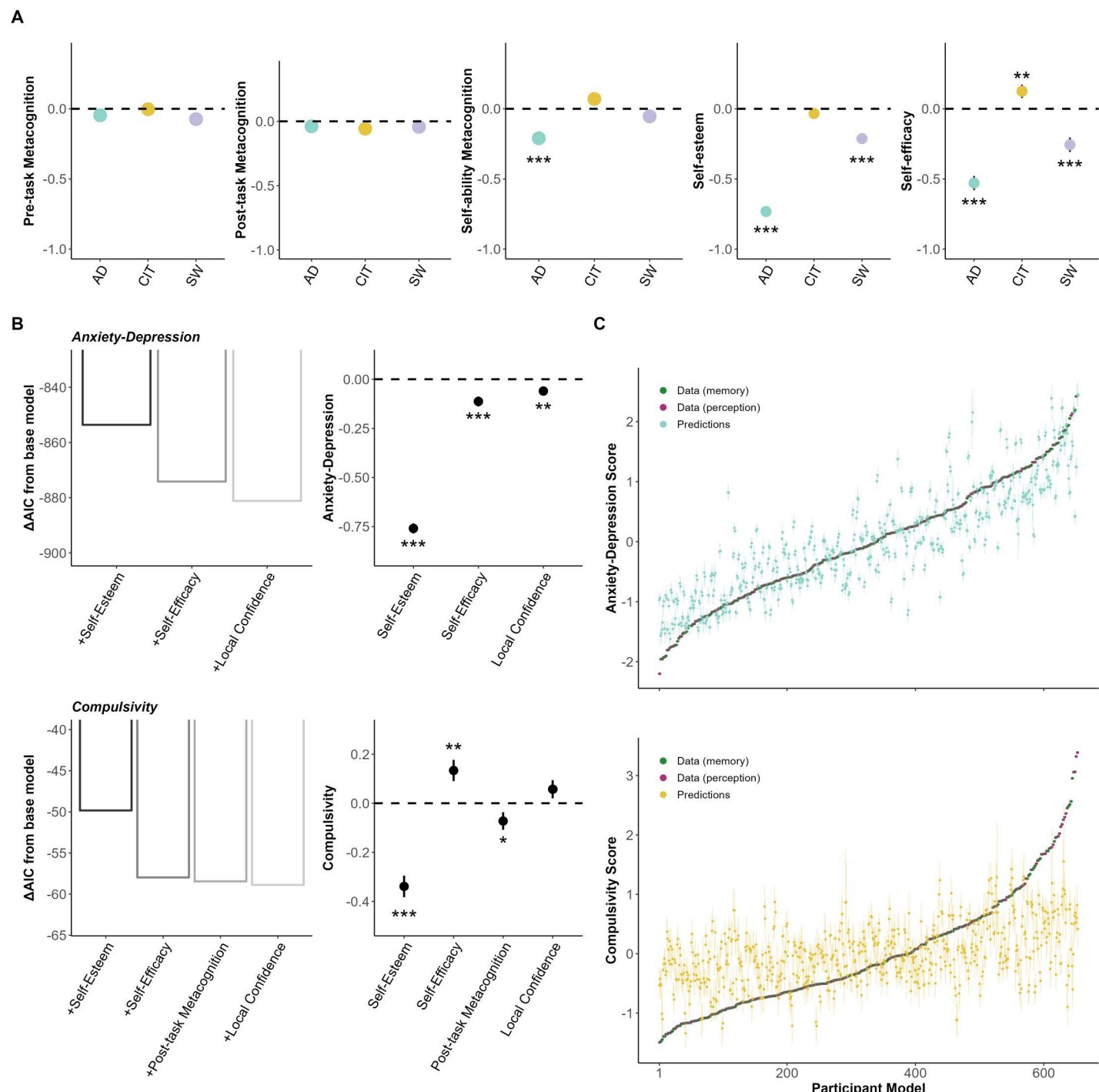

**Fig 4. Global metacognition and its association to dimension severity. (A)** Various measures of global metacognition and its association with dimension scores. Transdiagnostic dimension scores were included in the same model, with a separate model for each metacognitive measure. The Y-axes shows the z-scored change in metacognition level as a function of 1 standard deviation increase of dimension scores. Error bars denote standard errors. *p < 0.05, **p < 0.01, ***p < 0.001. **(B)** Stepwise regression of metacognitive measures contributing to psychopathology. Insets on the left depict the Akaike information criterion (AIC) scores from the models. Base model includes IQ, age and gender, and every additive iteration includes the regressor following rightward of the x-axes. The lower the AIC, the better the model fit. The right insets depict coefficients of the metacognitive regressors from the final model of the stepwise regression. Previously non-significant relationships in (A) (e.g., post-task metacognition and compulsivity) become significant when all relevant metacognitive metrics were included in the same model predicting dimension score. The Y-axes shows the change in dimension score as a function of 1 standard deviation increase of z-scored metacognitive regressor. Error bars denote standard errors. *p < 0.05, **p < 0.01, ***p < 0.001. **(C)** Out-of-sample model predictions from cross-validation of the winning model predicting dimension severity. Models are the same as the ones revealed by step-wise regression analyses. X-axis represents participant models, where there are two data confidence models

(purple for perception and green for memory) per participant, ranked by dimension score. Error bars indicate 95% confidence interval of predicted score. See S5 File for comparison of adjusted $R^2$ and root mean square error (RMSE) of all the tested models.

## Local and global metacognitive contributions to anxiety-depression and compulsivity differ

Lastly, a recent study has revealed differential associations between local (i.e., trial-by-trial (perceptual) task confidence ratings) and global (i.e., post-task confidence, self-esteem) measures of metacognition with dimension scores [22]. We extended this work by iteratively examining the importance of our six different local and global metacognitive metrics (across memory and perception) in predicting psychiatric dimensions.

We started with a forward step-wise regression approach, utilising a base model of IQ, age and gender to predict dimension scores. We observed the impact of iterative addition of local confidence, pre-task metacognition, post-task metacognition, self-ability metacognition, self-esteem and self-efficacy on the model fit (Fig 4B). For anxiety-depression, low self-esteem ($\beta = -0.76$, SE = 0.03, $p < 0.001$; $\Delta$AIC = −853.62) was the biggest contributor, followed by low self-efficacy ($\beta = -0.11$, SE = 0.03, $p < 0.001$; $\Delta$AIC = −20.53), and low local task confidence ($\beta = -0.06$, SE = 0.02, $p = 0.003$; $\Delta$AIC = −6.98). For compulsivity, the most impactful predictor was again low self-esteem ($\beta = -0.34$, SE = 0.04, $p < 0.001$; $\Delta$AIC = −49.83), then high self-efficacy ($\beta = 0.13$, SE = 0.04, $p = 0.002$; $\Delta$AIC = −8.14), low post-task metacognition ($\beta = -0.07$, SE = 0.04, $p = 0.04$; $\Delta$AIC = −0.48) and high local task confidence ($\beta = 0.06$, SE = 0.04, $p = 0.12$; $\Delta$AIC = −0.43). Interestingly, we replicated a bi-valenced contribution of different metacognitive levels to compulsivity—high local confidence and low global self-esteem both emerged as key predictors in perception [22]—and demonstrate that this same pattern is also obtained in a memory task, indicative of a domain-general effect. To ensure the robustness of the step-wise regression models, we validated them by running a repeated 5-fold cross-validation procedure on 16 model variations for comparison (see S5 File for details). We found the same results as the step-wise regression, with out-of-sample prediction of the best metacognitive model being highly predictive for anxiety-depression (r = 0.86, $p < 0.001$, 95% CI = [0.84 0.88]), and moderately predictive for compulsivity (r = 0.44, $p < 0.001$, 95% CI = [0.38 0.50]) (Fig 4C).

Overall, our findings suggest that variance in anxiety-depression and compulsivity is predominantly captured by the global factor of low self-esteem, with compulsivity also additionally sensitive to multiple and bi-valenced contributions of more local levels of metacognition.

## Discussion

In this study, we sought to reconcile diverging findings in how metacognition relates to psychiatric symptoms/subclinical traits, particularly when comparing more traditional clinical approaches to metamemory [11–16] with recent transdiagnostic methods applied to metaperception [21–24,26]. We considered three possible reasons for this divergence: (i) a metacognitive domain-specificity leading to distinct results with memory and perception tasks, (ii) the co-occurrence of mental health traits overshadowing effects, and/or (iii) differing manifestations of psychopathology at different hierarchical levels of metacognition. We were able to refute the first explanandum by showing that lower confidence is linked to anxiety-depression, and (partially) higher confidence is linked to compulsivity, across both perception and memory tasks. Secondly, using a dimensional approach, we replicated the opposing effects of anxiety-depression and compulsivity, lending support for the overshadowing hypothesis. Lastly, we found that mental health dimensions were differentially linked

to distinct levels of a metacognitive hierarchy, refining accounts of local versus global associations with mental health.

Whether metacognition is domain-general [31,32,36,52−57] or specific [27−32] is a subject of continued debate. We take the view that the extent of domain-generality is likely to differ between different metacognitive measures. For instance, we observed that metacognitive bias (average confidence level) was robustly correlated across domains, whereas metacognitive efficiency (the mapping between performance and confidence) was more weakly associated (S3 File), between perception and memory tasks. It is probable that metacognitive efficiency is influenced by other factors, such as interindividual differences in the utilisation of domain-specific cues (e.g., familiarity of images versus spatial information of dots) that determine the evaluative accuracy of confidence reports [58,59], and are thus more task-specific. Notably, our findings go beyond previous work on the domain-generality of metacognition in relation to mental health which focused on perception and semantic knowledge tasks [21] by showing that mappings between confidence and symptomology generalise across perception and (episodic/working) memory tasks—a domain central to our understanding of compulsivity/OCD. We saw that metacognitive bias was dissociatively linked to anxiety-depression (underconfidence) and compulsivity (overconfidence). Even though the evidence from our non-patient sample is indirect, it does support the notion that the higher co-morbid anxiety/depression symptoms commonly found in OCD patients might explain the reduction in memory confidence seen in clinical OCD studies [60] but which is accounted for when studying compulsive individuals using a dimensional approach [25,34]. However, recent studies with OCD patients have found lowered confidence in comparison to healthy controls despite adjusting for anxiety and depression scores [61,62]. It is thus also possible that there are specific features in patients that make them categorically distinct from "non-patients" [63] and lead to metacognition varying across domains differently between these two cohorts, or that other forms of co-morbid psychopathology could explain this lowered confidence. The true test will require a comparison with a transdiagnostically-characterised clinical cohort [20].

We note that the association between high confidence and compulsivity was not particularly strong in our data—the effect was small ($\beta = 0.07$ (CIT) versus $\beta = −0.12$ (AD)) and became non-significant when examined in the perceptual domain only (S6 File). We considered several explanations for this through control analyses, and determined that it was not due to differences between the tasks in the (i) accuracy titrated during the tutorial, (ii) pre-task metacognitive evaluations, (iii) task order completion, (iv) task accuracy titration, or due to (v) using transformed scores versus de novo factor analysis scores. Because a relationship between high perceptual confidence and compulsivity has been replicated numerous times [21,22,24,26], we suggest that our null finding might be driven by a slightly smaller effect size ($\beta\sim = 0.2$) than we expected in our power calculation (which led to our somewhat smaller sample size). Future studies might note the importance of ensuring higher power for investigating relationships between metacognition and psychopathology.

We unexpectedly observed that social withdrawal exhibited similar associations with metacognition as those found for anxiety-depression, which was never seen before in prior work [21,22,26]. Although we found that social anxiety questionnaire score distributions in the current sample were similar to those of our prior study [26], we also found that correlations between the anxiety-depression and social withdrawal factors were higher than expected ($r = 0.53$ versus $r = 0.43$ [26] or $r = 0.33$ [22]). Indeed, when social withdrawal was not included in the regression models, our anxiety-depression effects became much stronger (e.g., $\beta = −0.12$ to $\beta = −0.20$ when predicting local confidence). As social withdrawal has not shown significant relations with metacognition in any of the prior studies using the full three-factor

transdiagnostic approach [21,22,26] we refrain from interpreting these effects further and focus on anxiety-depression and compulsivity.

Recent studies have started to unravel distinct contributions of different levels of a metacognitive hierarchy to ageing [36] and mental health [22]. For the latter, differential contributions across local confidence (in perception), global confidence and self-beliefs to transdiagnostic dimensions were found. In this study, we go beyond a narrow focus on metaperception to include metamemory. Additionally, we comprehensively quantified six metacognitive metrics consisting of local task mean confidence and more global levels of metacognition—pre-task metacognition, post-task metacognition, cognitive self-ability metacognition, self-esteem and self-efficacy. We first observed that all metacognitive metrics we assessed were positively associated with each other (S7 File), but the associations between local and global levels were not particularly high (e.g., local confidence with any other metric, r = 0.13–0.33). Thus, it is unsurprising that these different levels of a metacognitive hierarchy showed different relationships with anxiety-depression and compulsivity. When we probed how much each metacognitive measure contributes to dimension severity, we found that anxiety-depression and compulsivity were linked to different yet overlapping profiles of hierarchical metacognition across step-wise and cross-validation approaches.

For anxiety-depression, a majority of the variance was explained by global metacognition in decreased self-esteem levels, followed by lower self-efficacy and lower local confidence. The predominant contribution of low global metacognition echoes clinical models of depression that emphasise negative schemas about general topics (oneself, the world, the future) [64,65]. For compulsivity, there was a contribution of lower self-esteem and lower post-task metacognition, but also higher self-efficacy and higher local confidence effects. This (replicated [22]) bi-valenced effect of different levels of a metacognitive hierarchy in compulsivity might encapsulate the seemingly contrasting hypotheses of OCD as a disorder of doubt [9,10,66] versus the rigidity of confidence and faulty world models underlying compulsive behaviour [25,67]. For the former, low self-esteem/post-task metacognition might reflect distrust of one's judgements/actions that promote repetitive behaviours like checking or assurance seeking [68]. For the latter, high local confidence/self-efficacy might reflect the greater endorsement of individuals' repetitive actions as being helpful despite evidence to the contrary [69,70]. These two phenomena are likely to act in tandem to aggravate persistent compulsive behaviours.

In summary, we found that metacognition exhibits similar patterns across perception and memory, but shows distinct, differing hierarchical contributions to psychopathology.

## Supporting information

**S1 File. Tutorial and task performance staircase titration.**
(PDF)

**S2 File. Harm versus non-harm stimuli effect on memory accuracy and confidence.**
(PDF)

**S3 File. Meta-d' model parameter analyses.**
(PDF)

**S4 File. Distributions and correlations of questionnaire and dimensional scores.**
(PDF)

**S5 File. Cross validation of regression models predicting dimension scores.**
(PDF)

**S6 File. Task domain specificity of metacognition and behaviour.**
(PDF)

**S7 File. Correlations between metacognitive measures across the hierarchy.**
(PDF)

## Acknowledgments

We thank Claire Gillan for their discussion on the initial analyses of the study.

## Author contributions

**Conceptualization:** Tricia X. F. Seow, Tobias U. Hauser.

**Data curation:** Tricia X. F. Seow.

**Formal analysis:** Tricia X. F. Seow.

**Funding acquisition:** Tricia X. F. Seow, Tobias U. Hauser.

**Investigation:** Tricia X. F. Seow.

**Methodology:** Tricia X. F. Seow, Stephen. M. Fleming, Tobias U. Hauser.

**Software:** Tricia X. F. Seow.

**Supervision:** Tobias U. Hauser.

**Writing – original draft:** Tricia X. F. Seow.

**Writing – review & editing:** Tricia X. F. Seow, Stephen. M. Fleming, Tobias U. Hauser.

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
