## [Decision Letter · Decision Letter 0]

24 Sep 2024

PMEN-D-24-00256

Metacognitive biases in anxious-depression and compulsivity extend across perception and memory

PLOS Mental Health

Dear Dr. Seow,

Thank you for submitting the manuscript you and your colleagues have prepared for potential publication in PLOS Mental Health ("Metacognitive biases in anxious-depression and compulsivity extend across perception and memory" (PMEN-D-24-00256).

I first would like to apologize for the long delay between your submission and the present message. I was finally able to complete the review process and have been fortunate to obtain two reviewers who possess expertise in the topic you investigated.

Based upon the reviews, and my own consideration of the paper, we feel that you work has merit but does not fully meet PLOS Mental Health’s publication criteria as it currently stands. Therefore, we invite you to submit a revised version of the manuscript that addresses the points raised during the review process.

---

As you will note upon consulting the comments, experts identified strengths in your work, expressed an overall positive evaluation of it, and acknowledged the importance of the topic. Whereas Reviewer#1 only has minor points to address,Reviewer#2 identified major issues, some quite serious, that require your attention .

You will find some of these points below for your consultation. However, please note that the integrality of the Reviewer #2's review is attached to this message as a pdf document (Review_2024-09-09).

Overall, Reviewer#2 identifies weaknesses and ambiguities in the argumentative flow of the manuscript and, among other things, requests that the appearance of new concepts all along the text be contextualised in a more rigorous manner. The reviewer also highlights a lack of justification for the statistical tools used so as a lack of details in the description of the analyses and the reporting of the results, which unfortunately blurs the relevance of the researchers' analytical strategy and casts doubt on the coherence of the observed effects.

---

On my part, I must also bring to your attention that the organization of your manuscript is not in line with the PLOS Mental Health submission guidelines , that you can find here: https://journals.plos.org/mentalhealth/s/submission-guidelines . I therefore encourage you to modify the manuscript accordingly.

----

---

I wish you all the best in the revision of your work.

Kind regards,

Pierre Olivier Jacquet, PhD

Academic Editor

PLOS Mental Health

---

Journal Requirements:

1. We ask that a manuscript source file is provided at Revision. Please upload your manuscript file as a .doc, .docx, .rtf or .tex.

Additional Editor Comments (if provided):

Reviewers' comments:

Reviewer's Responses to Questions

**Comments to the Author**

1. Does this manuscript meet PLOS Mental Health’s publication criteria ? Is the manuscript technically sound, and do the data support the conclusions? The manuscript must describe methodologically and ethically rigorous research with conclusions that are appropriately drawn based on the data presented.

Reviewer #1: Partly

Reviewer #2: Yes

2. Has the statistical analysis been performed appropriately and rigorously?

Reviewer #1: N/A

Reviewer #2: Yes

3. Have the authors made all data underlying the findings in their manuscript fully available (please refer to the Data Availability Statement at the start of the manuscript PDF file)?

Reviewer #1: Yes

Reviewer #2: Yes

4. Is the manuscript presented in an intelligible fashion and written in standard English?

Reviewer #1: Yes

Reviewer #2: Yes

5. Review Comments to the Author

Reviewer #1: 1. The methodology used can provide valuable insights with regard to the research questions. However, the paper lacks overall clarity and precision, especially with regard to its objectives and methodology description. These elements must be clarified to properly assess the appropriateness of the conclusions.

2. The analyses are not sufficiently detailed, and it is unclear which specific analysis was used to address each research question.

Reviewer #2: This paper addresses an apparent dissociation in the empirical observations regarding metacognitive processes in the general population when confronted with psychopathological dimension or “symptoms” derived from self-questionnaires: while anxious-depression traits are associated with diminished confidence in one’s cognitive performance in psychometric tasks, scoring high on compulsive and intrusive thoughts is associated with overconfidence in the same type of cognitive tasks. The study here proposed by Seow, Fleming and Hauser attempt to refine our understanding of such observations reported in previous papers of their groups and in those of other colleagues.

To do so, they contrast metacognitive performance metrics across two different tasks, a perceptual decision task vs. a working memory task, to generalize across domains. Testing those two tasks in the same participants (n=327) sampled from the general population thanks to online testing give the authors enough power to replicate earlier results across domains. It also sheds light on the aforementioned dissociation between the two psychopathological traits with respect to local (trial by trial) or global metacognitive assessments.

The strength of this report lays in the relatively large number of participants performing multiple well controlled tasks and responding to extensive questionnaires, all of which is analyzed through cogent statistical procedures. The authors deployed appropriate control procedure to ensure the quality of data collected online which is notoriously less reliable than that collected in the lab. The analyses are well-conducted. In general, the article is well constructed and easy to read.

I will have very few remarks: Authors, as in earlier publications of theirs and others, describe self-report questionnaire scores as “symptoms” (e.g. “anxious depression symptoms”). However, in the general population it would be more appropriate to speak of subclinical traits or psychopathological dimensions. Especially given, the fact that, as they themselves note that observations in psychiatric patients (e.g. OCD) may not be entirely consistent with the outcome observed in association to traits like “compulsive impulsive”. At least, the manuscript should explicitly acknowledge the slightly abusive use of the word “symptoms” when referring to psychological dimensions, defined in the general non-clinical population whose concurrent validity to actual psychopathological processes in patients remains to be demonstrated.

6. PLOS authors have the option to publish the peer review history of their article (what does this mean? ). If published, this will include your full peer review and any attached files.

**Do you want your identity to be public for this peer review?** For information about this choice, including consent withdrawal, please see our Privacy Policy .

Reviewer #1: No

Reviewer #2: No

---

## [Decision Letter · Decision Letter 1]

27 Jan 2025

Metacognitive biases in anxiety-depression and compulsivity extend across perception and memory

PMEN-D-24-00256R1

Dear Dr Seow,

We are pleased to inform you that your manuscript 'Metacognitive biases in anxiety-depression and compulsivity extend across perception and memory' has been provisionally accepted for publication in PLOS Mental Health.

Best regards,

Pierre Olivier Jacquet, PhD

Academic Editor

PLOS Mental Health

Dear Authors,

The reviewer has no objections to your revisions.

I am therefore pleased to announce the decision to accept your manuscript for publication in PLOS Mental Health.

I wish you all the best

Pierre O Jacquet

Reviewer Comments (if any, and for reference):

Reviewer's Responses to Questions

**Comments to the Author**

1. If the authors have adequately addressed your comments raised in a previous round of review and you feel that this manuscript is now acceptable for publication, you may indicate that here to bypass the “Comments to the Author” section, enter your conflict of interest statement in the “Confidential to Editor” section, and submit your "Accept" recommendation.

Reviewer #1: All comments have been addressed

2. Does this manuscript meet PLOS Mental Health’s publication criteria ? Is the manuscript technically sound, and do the data support the conclusions? The manuscript must describe methodologically and ethically rigorous research with conclusions that are appropriately drawn based on the data presented.

Reviewer #1: Yes

3. Has the statistical analysis been performed appropriately and rigorously?

Reviewer #1: Yes

4. Have the authors made all data underlying the findings in their manuscript fully available (please refer to the Data Availability Statement at the start of the manuscript PDF file)?

Reviewer #1: Yes

5. Is the manuscript presented in an intelligible fashion and written in standard English?

Reviewer #1: Yes

6. Review Comments to the Author

Reviewer #1: The authors have addressed all of my comments and concerns. I thank them for their thoughtful consideration.

7. PLOS authors have the option to publish the peer review history of their article (what does this mean? ). If published, this will include your full peer review and any attached files.

**Do you want your identity to be public for this peer review?** For information about this choice, including consent withdrawal, please see our Privacy Policy .

Reviewer #1: No
